# Utilization of AhR and GPR35 Receptor Ligands as Superfoods in Cancer Prevention for Individuals with IBD

**DOI:** 10.3390/ijms26189160

**Published:** 2025-09-19

**Authors:** Olga Poźniak, Robert Sitarz, Monika Zofia Sitarz, Dorota Kowalczuk, Emilia Słoń, Ewa Dudzińska

**Affiliations:** 1Department of Dietetics and Nutrition Education, Medical University of Lublin, 20-093 Lublin, Poland; olga.pozniak@umlub.pl (O.P.); ewa.dudzinska@umlub.pl (E.D.); 2Department of Human Anatomy, Medical University of Lublin, 20-093 Lublin, Poland; 3I Department of Surgical Oncology, St. John’s Cancer Center, 20-090 Lublin, Poland; 4Department of Conservative Dentistry with Endodontics, Medical University of Lublin, 20-090 Lublin, Poland; monikasitarz@umlub.pl; 5Chair and Department of Medicinal Chemistry, Medical University of Lublin, 20-090 Lublin, Poland; dorota.kowalczuk@umlub.pl (D.K.);

**Keywords:** chronic inflammation, inflammatory bowel disease, cytochrome P450, aryl hydrocarbon receptor, GPR35 receptor, colorectal cancer, tumorigenesis

## Abstract

Carcinogenesis is a complex process characterized by the uncontrolled proliferation of abnormal cells, influenced by environmental, genetic, and epigenetic factors. Chronic inflammation is undoubtedly one of the key contributors to carcinogenesis. Inflammatory bowel disease (IBD) is associated with an increased risk of colorectal cancer (CRC) due to persistent inflammation resulting from continuous immune system activation and excessive immune cell recruitment. IBD is also linked to certain nutritional deficiencies, primarily due to dietary modifications necessitated by the disease’s pathophysiology. Consequently, individualized nutritional supplementation appears to be a rational approach to addressing these deficiencies. The use of functional foods, including anti-inflammatory nutraceuticals, in individuals with IBD may play a crucial role in modulating cellular pathways that inhibit the release of inflammatory mediators. Thus, the regulation of the aryl hydrocarbon receptor (AhR) and G protein-coupled receptor 35 (GPR35) through dietary ligands appears to be of significant importance not only in the treatment of IBD and maintenance of remission but also in the prevention of tumorigenic transformation, particularly in genetically predisposed individuals. This narrative review was conducted using PubMed, Scopus, and Web of Science databases. The search covered literature published between January 2000 and June 2024. Keywords included ‘inflammatory bowel disease’, ‘colorectal cancer’, ‘AhR’, ‘aryl hydrocarbon receptor’, ‘GPR35’, ‘cytochrome P450’, ‘nutraceuticals’, ‘probiotics’, and ‘superfoods’. Only English-language articles were included. The selection focused on studies investigating mechanistic pathways and the role of dietary ligands in AhR and GPR35 activation in IBD and CRC. The SANRA guidelines for narrative reviews were followed to ensure transparency and minimize bias.

## 1. Introduction

As early as the 19th century, Rudolf Virchow proposed that chronic inflammation might contribute to cancer development, noting that tumors often emerge at sites of persistent inflammatory activity. He suggested that prolonged tissue irritation and inflammation could promote excessive cell proliferation and enhance the likelihood of malignant transformation [1].

The development of cancer is a multifactorial process involving the unregulated growth of atypical cells, shaped by environmental exposures, genetic predispositions, and epigenetic alterations [2]. While the interplay between inflammation and tumorigenesis is increasingly evident, the underlying molecular pathways and effective preventive interventions are not yet fully defined. Contemporary research underscores the involvement of immune components such as cytokines, immune cells, and inflammatory mediators, across all phases of tumor evolution, from initiation to metastasis [3]. Chronic inflammation, microbiome alterations, genetic factors, and dietary components interact in a complex network influencing colorectal cancer risk in IBD patients. The interplay between CYP enzymes, AhR and GPR35 signaling, and diet-derived ligands represents a potential target for innovative therapeutic and preventive strategies [3,4].

Sustained inflammatory responses, mediated by immune cells and key pro-inflammatory cytokines such as TNF-α, IL-6, and IL-1β, play a pivotal role in shaping the tumor immune microenvironment (TME) and facilitating cancer progression. These processes involve activation of signaling cascades including the NF-κB and MAPK pathways. Moreover, chronic inflammation fosters genomic instability and promotes the clonal selection of more malignant cell populations [2].

It is estimated that only a small fraction (approximately 5–10%) of cancer cases are attributable to inherited germline mutations, with the majority linked to modifiable factors such as diet, exposure to environmental toxins, and autoimmune diseases [5]. Within this framework, cytochrome P450 (CYP) enzymes play a pivotal role in the metabolism of numerous substances, both endogenous and xenobiotic, highlighting their importance in maintaining cellular homeostasis and influencing cancer risk [6].

The human body is continuously exposed to numerous chemical substances, including pharmaceuticals, cosmetics, environmental pollutants, pesticides, and harmful dietary components such as food additives. Collectively, these substances form a group of xenobiotics that may be potentially toxic and contribute to cancer development [6]. The oxidative activation of carcinogens by P450 enzymes leads to the formation of electrophilic reactive intermediates, which can bind to DNA, resulting in DNA adducts that may cause mutations [7].

These carcinogenic factors share common mechanisms, including the disruption of tissue homeostasis and the induction of a persistent protective response—chronic inflammation. An inflammatory microenvironment rich in immune cells, cytokines, and DNA damage induces various epigenetic changes, while the accumulation of mutations in adjacent epithelial cells triggers pro-carcinogenic effects such as immunosuppression and angiogenesis, thereby promoting sustained cell proliferation and tumor initiation [5].

Once a tumor has formed, it can perpetuate chronic inflammation by releasing pro-inflammatory factors and recruiting immunosuppressive immune cells, thereby further altering the tumor immune microenvironment (TME). This process suppresses the anti-tumor immune response by inhibiting antigen presentation and cytotoxic T-cell activation, ultimately promoting tumor progression. Thus, the bidirectional interaction between chronic inflammation and the TME is pivotal and plays a central role in carcinogenesis and tumor development [2].

One of the most well-documented examples of inflammation-driven carcinogenesis is inflammatory bowel disease (IBD) [8]. It is now widely recognized that patients with IBD particularly those with ulcerative colitis (UC), and to a lesser extent Crohn’s disease (CD) are at an increased risk of developing cancer. The type of malignancy that arises in the context of IBD is considered a representative model of inflammation-associated tumorigenesis [2].

## 2. IBD—Pathological and Clinical Features and Its Association with Carcinogenesis

IBD is primarily classified into two major subtypes: ulcerative colitis and Crohn’s disease. In UC, the inflammatory process is confined to the mucosal and submucosal layers of the colon, whereas in CD, inflammation can affect any part of the gastrointestinal tract and involves all layers of the intestinal wall. The ileum is the most commonly affected site in CD [9].

The pathogenesis of IBD results from a complex interaction between environmental factors, the immune system, susceptibility genes, and alterations in the host microbiome. This interplay leads to excessive immune system activation, characterized by infiltration of the lamina propria with lymphocytes, macrophages, neutrophils, and other immune response cells. The role of inflammatory cells in sustaining active disease is well established, and most therapeutic approaches aim to inhibit the cascade of pro-inflammatory cytokines [10].

Currently, there is no definitive cure for IBD, although some systemic treatments—such as anti-TNF-α therapy—have shown promising effects. However, these treatments are often associated with significant adverse effects [10].

IBD is most commonly diagnosed in individuals aged 20 to 40 years, although onset can occur at any age. It is a chronic condition characterized by alternating periods of clinical relapse and remission [11,12].

Patients with IBD have an increased risk of developing colorectal cancer (CRC) compared with the general population. Notably, the incidence rates of IBD-associated CRC (IBD-CRC) have declined in recent decades [13]. Recent studies indicate that the risk of developing CRC in UC patients is approximately 2% after 10 years and up to 18% after 30 years of disease duration [14].

The carcinogenic pathway in IBD-associated CRC differs from that of sporadic CRC. While sporadic CRC follows the adenoma–carcinoma sequence, IBD-CRC develops through an inflammation–dysplasia–carcinoma sequence with an underlying immune-mediated mechanism [15].

Additionally, recent findings suggest that patients with IBD are at a higher risk of developing extraintestinal malignancies, likely due to prolonged immunosuppressive therapy and chronic inflammation [16]. Immunosuppressive therapies used in IBD, such as TNF-α inhibitors and corticosteroids, may interact with AhR and GPR35 pathways either directly or indirectly through the modulation of microbiota composition. Understanding these interactions is crucial for personalized treatment approaches [17,18].

Several risk factors for IBD-CRC have been identified, including disease duration, extent of inflammation, and primary sclerosing cholangitis. Although data remain limited, early-onset IBD and pediatric cases appear to be associated with a higher incidence of CRC. Unlike sporadic CRC (sCRC), IBD-CRC is driven by prolonged chronic inflammation, which initiates and promotes tumorigenesis [19].

Emerging research has identified additional factors involved in IBD-related carcinogenesis. One key enzyme under investigation is cytochrome P450 1A1 (CYP1A1), a major member of the cytochrome P450 family, activated via the aryl hydrocarbon receptor (AhR). The AhR pathway plays a crucial role in various cellular signaling mechanisms that regulate the cell cycle and maintain homeostasis. Dysregulation of these pathways has been linked to tumor progression. Additionally, CYP1A1 is involved in both the detoxification of environmental carcinogens and the metabolic activation of dietary compounds with potential cancer-preventive effects [20].

Therefore, the dietary regulation of AhR ligands appears to be of significant importance—not only in preventing cancer development in IBD patients but also in managing IBD itself.

## 3. The Significance of CYP1A1 and CYP1B1 Gene Expression in Inflammatory Bowel Diseases and Colorectal Cancer

The cytochrome P450 enzyme system is responsible for the phase I metabolism of xenobiotics, including numerous toxic compounds that enter the human body through industrialization and environmental pollution, such as food contaminants [21]. Therefore, both CYP gene polymorphisms and environmental factors influence the activity of CYP450 enzymes by modulating their function—either by inducing or inhibiting them [22]. Studies indicate that both CYP1A1 and CYP1B1 isoforms of cytochrome P450 are expressed in the gastrointestinal tract [23,24].

The expression of CYP1A1 and CYP1B1 genes is induced and regulated by AhR, which resides in the cytosol in an inactive form, bound to multiple chaperone proteins. Upon ligand binding, AhR is released and forms a dimer with the nuclear translocator ARNT. The AhR/ARNT complex subsequently activates the transcription of target genes, including cytochrome P450 genes such as CYP1A1 and CYP1B1 [25].

This heterodimer regulates gene expression by binding to genomic DNA at specific regions known as AhR response elements (AHREs; 5′-GCGTG-3′), also referred to as xenobiotic response elements or dioxin response elements. AhR regulates various prototypical genes, including CYP1A1, CYP1A2, and CYP1B1 [26,27]. Additionally, AhR controls the expression of numerous other genes involved in diverse signaling pathways. Prolonged activation of AhR by an agonist not only initiates gene transcription but also induces the synthesis of the AhR repressor protein (AhRR). AhRR, located in the cell nucleus, binds to ARNT and forms an AhRR/ARNT heterodimer, which competes with the AhR/ARNT complex for binding to DRE sequences, thereby inhibiting AhR function [28] (Figure 1).

The primary ligands of the AhR are polycyclic aromatic hydrocarbons (PAHs) and organic chemical compounds from the dioxin group, which exhibit highly toxic properties. The role of CYP1A1 and CYP1B1 enzymes is to transform these compounds into more water-soluble forms, facilitating their excretion. However, in certain cases, these metabolic reactions may enhance the reactivity or biological activity of these compounds, leading to the formation of DNA adducts and genetic mutations [29].

Phase I enzymes can metabolically activate carcinogenic agents, generating genotoxic electrophilic intermediates. It is believed that the relative activity of these metabolizing enzymes—largely genetically determined—plays a crucial role in influencing cancer susceptibility in the host [30].

CYP1A1 and CYP1B1, therefore, catalyze the oxidation of procarcinogens into carcinogenic reactive intermediates. Consequently, the expression of these enzymes is a critical factor in carcinogenesis. Several studies have demonstrated differential expressions of CYP1A1 and CYP1B1 across various tumor types compared with normal tissue, suggesting their potential application in cancer prognosis and therapy [23].

Since CYP1B1 gene expression is influenced by exogenous activation of the AhR pathway by PAHs and related environmental pollutants, its expression in cancer patients may vary depending on lifestyle and environmental exposures [23]. Moreover, several studies have explored the association between CYP1B1 polymorphisms—Leu432Val (rs1056836), Asn453Ser (rs1800440), and Arg48Gly (rs10012)—and colorectal cancer risk, although findings remain inconclusive [30].

CYP1B1 has also been shown to metabolize endogenous compounds, such as arachidonic acid, producing hydroxyeicosatetraenoic acids, which may play a role in the development of inflammation, including IBD [31].

With respect to CYP1A1, studies have demonstrated that increased expression can lead to the accumulation of reactive metabolites, potentially disrupting immune responses in the gut and causing irreversible mucosal barrier damage. Conversely, CYP1A1 expression in enterocytes of patients with UC has been reported to be lower than in healthy control groups [24]. Other evidence suggests that CYP1A1 may play a protective role in regulating the intestinal immune response. However, certain CYP1A1 gene polymorphisms may contribute to IBD development [24]. Polymorphisms in CYP1A1 and CYP1B1 genes can affect enzymatic activity, influencing the metabolism of AhR ligands and xenobiotics. Some studies suggest an association between specific variants and increased colorectal cancer risk in IBD, although findings remain inconsistent and require further large-scale studies [24].

Dioxins and other environmental toxins significantly enhance the generation of reactive oxygen species (ROS) in human cells—a process closely linked to AhR activation and the expression of CYP1A1 and CYP1B1. Given that AhR directly regulates these genes through its ligands, the use of specific AhR ligands as potential therapeutic agents for IBD, and for the prognosis and treatment of colorectal cancer, represents a promising direction for future research [32,33].

## 4. The Role of the AhR Receptor in Mitigating Inflammation in IBD

The aryl hydrocarbon receptor is a ligand-dependent transcription factor known for its ability to suppress inflammatory responses and attenuate experimental colitis. However, the precise mechanisms underlying its therapeutic potential in inflammatory bowel disease remain incompletely understood [25,27,34]. AhR plays a crucial role in maintaining the barrier between the host and the external environment. Activation of AhR by its ligands is essential for the proper development of innate intestinal immunity, particularly in early life. The gut microbiome contributes to this process by producing various AhR ligands, and disruption of this interaction may lead to the onset of colitis [35].

The environment exposes individuals to diverse exogenous AhR ligands, including polycyclic aromatic hydrocarbons (PAHs) found in tobacco smoke and charred foods such as grilled meat [36]. Endogenous AhR agonists include eicosanoids, bilirubin, indoxyl sulfate, and tryptophan metabolites. The AhR–ligand complex modulates inflammatory responses, regulates cell proliferation, and influences immune system activity, notably through the differentiation of T cells into Th17 cells [27].

Several studies have highlighted AhR’s role in regulating regulatory T cells (Tregs) and effector T cells by modulating the production of interleukin-10 (IL-10) and interleukin-22 (IL-22) [27]. Notably, IL-22 has been associated with intestinal mucosal healing, a key therapeutic goal in IBD management. A clinical study demonstrated that Indigo naturalis, an AhR ligand that activates the IL-22 pathway, exerts a strong therapeutic effect in UC. However, prolonged activation of IL-22 signaling may increase the risk of colitis-associated cancer [37].

In a preclinical study, mice were administered 3% dextran sodium sulfate (DSS) to induce colitis and treated with an AhR agonist. The results indicated that AhR deficiency increased susceptibility to colitis, while AhR activation using FICZ (a known agonist) mitigated DSS-induced colitis via the MK2/p-MK2/TTP pathway [25]. These findings suggest that AhR participates in inflammatory feedback loops contributing to IBD progression [36].

Champion et al. further demonstrated that pro-inflammatory cytokine stimulation increases AhR expression in Caco-2 intestinal epithelial cells through the NF-κB signaling pathway. Collectively, this evidence supports AhR as a promising therapeutic target in IBD, and points to the dietary modulation of AhR signaling as a potential prevention and treatment strategy [36].

Despite these advances, the identity and function of physiological AhR ligands remain incompletely defined [38]. Known exogenous ligands include dioxin and benzo[a]pyrene [39], but also natural compounds such as flavonoids, indoles, and carotenoids found in tea, herbs, fruits, and vegetables [39]. Examples include quercetin and curcumin [40].

The role of quercetin in AhR activation remains ambiguous. In MCF-7 breast cancer cells, quercetin functions as an AhR antagonist [41], whereas in HepG2 liver cells, it can induce AhR activation and stimulate CYP1A1 gene expression following 24 h of exposure [42]. Niestroy et al., found that quercetin promotes AhR-mediated responses in Caco-2 cells, likely due to its structural resemblance to both flavonoids and PAHs [43].

The search for endogenous ligands is ongoing, with arachidonic acid identified as a candidate [44]. Current research highlights the impact of dietary polyunsaturated fatty acids (PUFAs)—particularly n-3 and n-6 types—on inflammatory pathways. Arachidonic acid, derived from n-6 PUFA metabolism, fuels the arachidonic acid cascade, generating pro-inflammatory mediators such as prostaglandins, leukotrienes, thromboxanes, and reactive oxygen species [45]. It also enhances expression of TNF-α receptors on neutrophils, further promoting inflammation [45,46].

Tryptophan and its metabolites serve as crucial modulators of immune and inflammatory responses through AhR activation. Among them, kynurenic acid (KYNA) is a well-established endogenous AhR ligand [38]. KYNA’s affinity for AhR lies in the low micromolar range. However, in inflammatory and neo-plastic conditions, elevated KYNA levels are sufficient to activate AhR signaling [47,48].

Interestingly, AhR signaling and the abundance of known AhR ligands are significantly reduced in IBD patients, emphasizing the clinical importance of the AhR pathway in maintaining gut immune homeostasis [49].

## 5. Role—GPR35 and KYNA in IBD

Numerous studies highlight that genetic variants in individuals with IBD disrupt host–microbe interactions and compromise intestinal homeostasis by altering signaling pathways mediated by bacterial ligands and metabolites [50]. While current IBD treatments largely target cytokine production, growing evidence suggests that dietary and microbial metabolites can activate G protein-coupled receptors (GPCRs), many of which are involved in anti-inflammatory signaling [51].

One such receptor is GPR35, a GPCR that has been associated with IBD pathogenesis. Variants in the GPR35 gene have been linked to increased risk for CD and UC [52]. GPCRs are a large family of integral membrane proteins that mediate the conversion of extracellular stimuli into intracellular responses. Although they share a common seven-transmembrane helical structure, GPCRs can initiate diverse signaling cascades via both G protein-dependent and -independent pathways [53].

GPCRs are highly abundant in mammalian systems and are crucial for regulating numerous physiological processes in the intestine [54]. These receptors respond to a wide array of ligands—including lipids, peptides, and amino acid metabolites [55]. Upon ligand binding, GPCRs undergo a conformational change that facilitates G protein activation, initiating downstream signaling pathways that modulate inflammation, wound healing, angiogenesis, fibrogenesis, leukocyte trafficking, and even oncogenesis [55].

GPR35, an orphan GPCR, is predominantly expressed in intestinal epithelial cells (enterocytes) and certain immune cell subsets [56]. Wang et al., confirmed GPR35 expression in immune tissues and gastrointestinal organs, suggesting a key role in gut immune regulation [57]. Furthermore, genome-wide association studies (GWASs) have identified a T108M missense variant in the GPR35 gene as a risk factor for both CD and UC, reinforcing the receptor’s relevance in maintaining intestinal homeostasis [54].

## 6. Kynurenic Acid as a Key Ligand of the GPR35 Receptor

Kynurenic acid (KYNA), a metabolite of tryptophan, is a well-established endogenous ligand for GPR35. Other natural ligands include 2-oleoyl lysophosphatidic acid, while synthetic agonists include zaprinast, pamoic acid, and YE120 [54,58]. Given KYNA’s dual role as a ligand for both AhR and GPR35, it may act as a key molecular link between metabolic signaling and immune modulation in IBD.

Tryptophan (Trp) is an essential exogenous amino acid required for the proper functioning of the human body and must be obtained exclusively from dietary sources [59] (Figure 2).

Tryptophan is primarily found in animal-derived products such as pork, beef, poultry, lamb, and dairy, as well as in seeds, nuts, legumes, and whole grains [60].

Tryptophan plays a crucial role in various pathophysiological processes within the human body [59]. It supports the gut microbiota, intestinal epithelium, and immune cells [61], making it a key regulator of intestinal inflammation [62]. Studies have shown that Trp deficiency contributes to gut dysbiosis, which can lead to gastrointestinal inflammation and exacerbate IBD [62,63].

The gut microbiota is essential for generating natural ligands for the aryl hydrocarbon receptor (AhR), including Trp metabolites. Dysbiosis can reduce the production of these protective ligands, increase intestinal permeability, and shift signaling toward a pro-inflammatory state, which is implicated in carcinogenesis in IBD [62,63]. Maintaining Trp homeostasis may therefore be important for intestinal health.

Trp levels in the human body primarily depend on dietary intake and the activity of three major metabolic pathways in the gut [55,58]: the serotonergic pathway, which produces serotonin in enterochromaffin cells; the kynurenine pathway, active in immune cells, epithelial cells, and the lamina propria; and the indole pathway, which generates indole and its derivatives [57,60]. Each pathway contributes to host metabolism and the generation of bioactive metabolites that can modulate AhR signaling and other immune-related pathways.

The metabolism of kynurenine into kynurenic acid is facilitated by four kynurenine aminotransferases (KAT I, II, III, and IV), which catalyze the conversion of Kyn to KYNA [59].

KYNA exhibits species-dependent differences in potency, which is particularly low (median effective concentration (EC50) > 10^−3^ M) for human GPR35 [64,65]. Given that physiological KYNA concentrations in the human gut are generally lower than those required for effective GPR35 activation (EC50), supplementation or dietary enhancement may be necessary to achieve therapeutic effects [66,67,68]. The L-kynurenine metabolite KYNA is able to promote activation of GPR35, although the level of kynurenic acid reported to occur in the gut and the concentration required to activate GPR35 do not appear to overlap, at least in humans. In pharmacological assays, the addition of kynurenic acid was found to mobilize intracellular calcium concentration ([Ca^2+^]_i_) following the transfection of GPR35 alongside a cocktail of Gαq-based G protein chimeras, with an EC50 of 39 μM in humans, 11 μM in mice, and 7 μM in rats. Thus, a degree of species selectivity was observed between human and rodent orthologs of GPR35, with kynurenic acid displaying higher potency in rats than in humans [65].

KYNA signals its functions through the activation of GPR35, which is mainly expressed in immune cells and in the gastrointestinal tract, and may also participate in nociceptive perception. KYNA may exert anti-inflammatory effects through inhibition of tumor TNF-α induced by lipopolysaccharides in peripheral blood mononuclear cells [69].

Recent reports have shown that, in the serum of Crohn’s disease patients experiencing exacerbation, the levels of kynurenine and the kynurenine/tryptophan ratio were lower than in patients in remission, while tryptophan levels were increased [67,68].

Similarly, Małaczewska et al., reported that KYNA has immunotropic properties. They observed a pronounced increase in KYNA levels during infections and inflammation due to increased tryptophan breakdown along the kynurenine pathway, stimulated by inflammatory mediators such as pro-inflammatory cytokines or bacterial lipopolysaccharides [69].

Kynurenic acid is the most important agonist of the orphan G protein-coupled receptor GPR35, whose expression is observed in various types of immune cells [69].

The significant expression of GPR35 in immune cells and the increase in kynurenic acid concentration during inflammation suggest that this ligand–receptor pair may play an important role in immune regulation. Kynurenic acid has been observed to inhibit monocyte secretion of LPS-induced TNF-α in peripheral blood. This occurs because metabolic activation of the tryptophan pathway by pro-inflammatory stimuli also exerts an anti-inflammatory effect by increasing KYNA production. This provides an interesting feedback mechanism for immune response modulation [69].

Other studies have pointed out that a large quantity of GPR35 receptors is present in the crypts of Lieberkühn, which are rich in actively dividing stem and progenitor cells crucial for gastrointestinal epithelium self-renewal. An elevated plasma concentration of kynurenic acid has been observed in patients with ulcerative colitis or Crohn’s disease, potentially as a result of immune response modulation [58,69].

Kynurenic acid may exert an inhibitory effect on the phosphoinositide 3-kinase (PI3K)/protein kinase B (Akt) and mitogen-activated protein kinase (MAPK) pathways, which play crucial roles in generating inflammatory responses [47].

Moreover, there is evidence that KYNA–GPR35-mediated inhibition of adenylate cyclase contributes to the decrease in IL-23–IL-17 immune axis activity observed after KYNA treatment [47].

Additionally, a study by Walczak et al. demonstrated that KYNA inhibits the activation of Akt, extracellular signal-regulated kinases 1/2 (ERK1/2), and p38 kinase [70].

In contrast, Bishnupuri et al., reported that kynurenic acid inhibited Akt activation in HT29 cells and reduced cancer cell proliferation. These findings confirm that KYNA may exert chemo-preventive effects in colorectal cancer (CRC) by attenuating PI3K/Akt signaling and stabilizing β-catenin [71] (Figure 3).

Thus, kynurenic acid appears to have significant anti-inflammatory and, consequently, anticancer potential, as chronic inflammation is a well-recognized factor in carcinogenesis [72].

## 7. Nutraceuticals and Functional Foods Activating Anti-Inflammatory Pathways via AhR and GPR35 Receptor Activation as Potential Therapeutic Strategies for IBD

The incorporation of nutraceuticals into daily diet may contribute to improved overall health and physiological function. Nutraceuticals added to food products are classified as functional foods, a concept that is gaining increasing popularity [73].

Functional food refers to either naturally occurring foods or food products enriched with one or more additional compounds that may enhance physiological performance across all age groups or within specific demographic categories. Superfoods are defined scientifically as foods with a high nutrient density, rich in bioactive compounds that exert proven beneficial effects on health. 

The concept of ‘superfoods’ has emerged to describe foods with exceptionally high nutrient density and health-promoting properties [74]. Scientifically, superfoods are defined as foods rich in bioactive compounds with clinically proven effects on human health [74,75].

In the context of IBD and colorectal cancer prevention, this includes nutraceuticals such as kynurenic acid, quercetin, curcumin, and probiotics, which modulate AhR and GPR35 signaling pathways [74,75,76,77]. In this review, ‘superfoods’ are defined as natural foods or compounds with high concentrations of bioactive components that provide health benefits beyond basic nutrition, particularly by modulating inflammatory and metabolic pathways relevant to IBD and CRC.

Furthermore, studies have demonstrated the health benefits of functional foods produced through the incorporation of bioactive substances, such as nutraceuticals derived from plants, fruits, vegetables, and other natural sources. These compounds may exert positive effects on human health, including antioxidant, anti-inflammatory, antimicrobial, and anticancer activities [74,75,76].

## 8. Kynurenic Acid

Some studies suggest that kynurenic acid (KYNA) is being considered as a potential nutraceutical and may, in the future, be incorporated into food products [76].

The highest concentrations of KYNA have been found in bee products, broccoli, and certain varieties of potatoes. Additionally, some medicinal herbs, such as St. John’s wort (Hypericum perforatum), contain high levels of KYNA, highlighting their therapeutic potential for the gastrointestinal system [77,78].

Beyond dietary sources or endogenous synthesis via the kynurenine pathway in the gastrointestinal tract, gut microbiota possess aspartate aminotransferase—an enzyme analogous to mitochondrial KAT4—which catalyzes the transamination of kynurenine (KYN) to produce KYNA [77,79].

Alves et al. emphasize the potential nutraceutical value of kynurenic acid. Due to its limited bioavailability from food sources, exogenous administration as a nutraceutical supplement could enhance its physiological effects by modulating multiple cellular signaling pathways [4,73].

Dysregulation of these pathways has been associated with intestinal carcinogenesis and cancer progression [70]. Previous studies have indicated that KYNA exhibits anti-inflammatory, anti-ulcerative, and antioxidant properties [72].

KYNA is also being considered as a potential chemo-preventive agent for colorectal cancer (CRC), particularly in individuals with a familial predisposition. It may help reduce cancer risk or serve as an adjuvant in standard chemotherapy regimens. Preclinical studies have shown that intravenous administration of KYNA at doses of 50 or 100 mg/kg/h in rats was well tolerated [70].

However, other studies caution against the potential pro-carcinogenic effects of KYNA as an AhR ligand and highlight possible interactions between KYNA and GPR35 in cancer. Therefore, further research is required to clarify KYNA’s role in carcinogenesis.

Nevertheless, its anti-inflammatory properties suggest a degree of protective potential against cancer development [72].

AhR ligands such as KYNA and quercetin exhibit context-dependent effects. While low concentrations may promote intestinal barrier integrity and anti-inflammatory responses, higher doses or chronic exposure can enhance pro-carcinogenic pathways, highlighting the need for careful dose regulation [72].

## 9. Quercetin

Another important nutraceutical with anti-inflammatory potential is quercetin.

Quercetin (3,3′,4′,5,7-pentahydroxy-2-phenylchromen-4-one) is a representative of the flavanol subclass of flavonoids. This yellow pigment is found in various fruits and vegetables, including berries, cranberries, apples, onions, lovage, dill, and coriander. It is completely soluble in lipids and alcohol, poorly soluble in hot water, and insoluble in cold water [74]. It is primarily metabolized in the intestines and liver [80,81].

Quercetin has been described as a long-acting anti-inflammatory compound, with effects demonstrated in both animal models and human cell studies. It possesses mast cell-stabilizing and cytoprotective properties, particularly in the gastrointestinal tract.

Additionally, it modulates inflammation and immune responses [82] by reducing the production of pro-inflammatory cytokines and leukotrienes, inhibiting histamine release, and suppressing interleukin-4 (IL-4) production [81].

Quercetin may also enhance the efficacy of other drugs, including anticancer agents, by acting synergistically or reducing drug-related toxicity. Moreover, it is widely available as a standalone dietary supplement [80].

Many plant-derived flavonoids, including quercetin, are valued for their anti-inflammatory, antioxidant, anticancer, and antimicrobial properties [80,83].

Notably, quercetin is a common ingredient in dietary supplements, typically available in powder and capsule form, and is included in several anti-allergic medications [84].

## 10. Turmeric

Another significant bioactive compound with potential anticancer applications is curcumin. Curcumin, a polyphenol derived from the turmeric plant (*Curcuma longa* L.), is commonly used as both a spice and food coloring agent and has been widely utilized in traditional medicine [79]. It is recognized for its potent antimicrobial and immunomodulatory effects. Curcumin is considered safe for consumption—even at relatively high doses—as confirmed by toxicity studies. Due to its broad biological activity and low toxicity, it has been extensively investigated as a nutraceutical component in functional foods [76].

In addition to its anti-inflammatory and antioxidant effects, curcumin exhibits various health-promoting properties, including anticancer and anti-atherosclerotic activities [85].

The use of dietary supplements such as curcumins may be highly beneficial for strengthening the immune system and preventing diseases, including cancer. It represents a promising functional food option for individuals with compromised immunity, such as hospitalized patients and residents of long-term care facilities.

The incorporation of functional foods, including key nutraceuticals, is vital at every stage of human growth and development. The body requires a continuous supply of essential nutrients, especially during specific physiological conditions such as advanced age, severe illness-related malnutrition, and stress-induced immune suppression [76].

## 11. Lactobacillus

Studies have shown that Lactobacillus strains participate in tryptophan metabolism, leading to the production of AhR ligands. These ligands contribute to maintaining intestinal immune homeostasis and reducing the risk of inflammatory processes [86].

Certain Lactobacillus strains exhibit high capacities for AhR ligand production, supporting metabolic improvements and enhancing intestinal barrier function. One study found that activation of the AhR pathway by a potent AhR antagonist suppressed colonic inflammation. Interestingly, Lactobacillus strains demonstrate similar AhR-activating properties, effectively mimicking the anti-inflammatory effects of AhR antagonists [87].

A review by De la Rosa González et al., published in Foods, provides valuable insight into the role of probiotics—including *Lactobacillus* spp.—in AhR pathway signaling under both physiological and pathological conditions of the gastrointestinal tract. In healthy individuals, probiotics increase the production of AhR ligands, thereby enhancing intestinal immune tolerance. Under pathological conditions, especially in inflammatory states of the gastrointestinal tract, probiotic strains—particularly *Lactobacillus*—activate the AhR pathway primarily through tryptophan metabolites and short-chain fatty acid (SCFA) production, contributing to anti-inflammatory effects [87].

Another review by Huang et al. explores the mechanisms through which *Lactobacillus* spp., particularly *Lactobacillus reuteri*, influence immune responses. These bacteria secrete enzymes that convert tryptophan into AhR ligands, thereby activating the AhR signaling pathway. Given its central role in modulating immune responses, AhR activation offers compelling evidence for the therapeutic potential of probiotics in immune-mediated diseases. Notably, *L. reuteri* and related Lactobacillus species have demonstrated the ability to regulate inflammation in conditions such as colitis, celiac disease, endometritis, multiple sclerosis, type 1 diabetes, and ulcerative colitis. As such, targeting AhR activation via *L. reuteri* may represent a promising research avenue for developing treatments for various inflammatory and autoimmune diseases [88].

*Lactobacillus* strains are available not only as dietary supplements in capsule, tablet, and suspension forms but also in a variety of commonly consumed fermented foods, including yogurt, buttermilk, kefir, acidophilic milk, sauerkraut, and pickled cucumbers. In addition, certain fruit and vegetable ferments, bread, and sausages may be enriched with probiotic bacteria, depending on the manufacturer’s production methods [87,88,89].

Incorporating naturally fermented foods into daily diet may offer supportive benefits for both the treatment and prevention of inflammatory bowel diseases. These benefits may, at least in part, be mediated through modulation of key cellular pathways such as the aryl hydrocarbon receptor (AhR) and G protein-coupled receptor 35 (GPR35), which are involved in controlling inflammation and may contribute to tumor prevention, particularly in genetically predisposed individuals.

However, most evidence for KYNA, quercetin, curcumin, and probiotics comes from preclinical studies, and clinical validation remains limited. It is essential to distinguish between preclinical findings, early-phase clinical trials, and speculative hypotheses when discussing their potential roles in IBD and CRC.

## 12. Conclusions

This review synthesizes current evidence on the interplay between the aryl hydrocarbon receptor (AhR), G protein-coupled receptor 35 (GPR35), cytochrome P450 enzymes, and dietary ligands in the context of inflammatory bowel disease (IBD) and colorectal cancer (CRC). Emerging data suggest that certain dietary components—particularly tryptophan metabolites, polyphenols, and probiotics—may modulate these pathways, potentially contributing to intestinal homeostasis and attenuating chronic inflammation.

However, most evidence remains preclinical, derived from in vitro experiments and animal models, with limited clinical validation. The complexity of AhR and GPR35 signaling, their context-dependent effects, and inter-individual variability—due to genetics, microbiome composition, and concurrent therapies—underscore the need for caution when extrapolating these findings to clinical recommendations. 

Future research should prioritize well-designed clinical trials, standardized methodologies for dietary intervention studies, and the identification of biomarkers to monitor both efficacy and safety of AhR/GPR35-targeting strategies, to minimize potential carcinogenic risks, including the development of selective AhR modulators, dose optimization, and biomarker-based monitoring, to ensure safety in future clinical applications. Until such data are available, the dietary modulation of these pathways should be regarded as a promising but still experimental approach rather than a standard clinical recommendation.

## Figures and Tables

**Figure 1 ijms-26-09160-f001:**
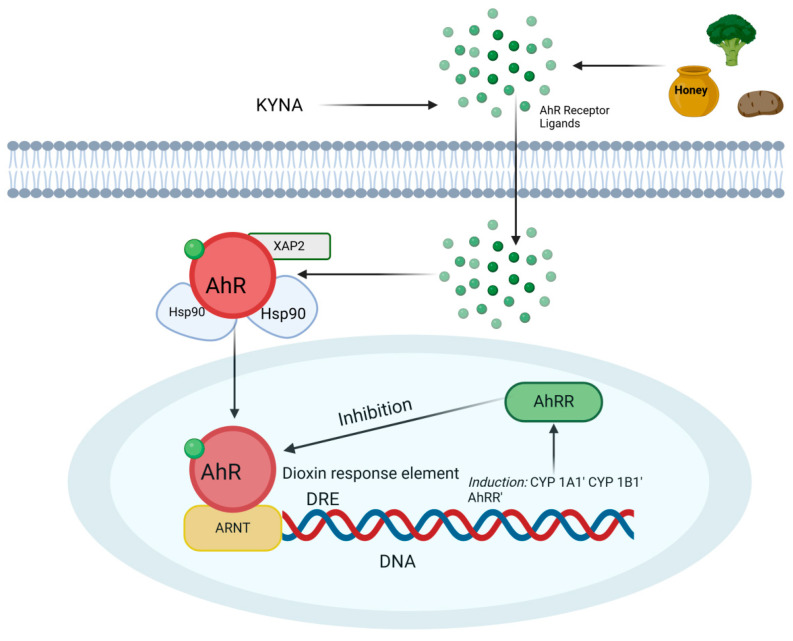
Schematic representation of CYP1A1 and CYP1B1 gene expression following activation of the aryl hydrocarbon receptor (AhR) by the tryptophan metabolite kynurenic acid (KYNA). In the cytoplasm, AhR is bound to a complex that includes heat shock protein 90 (HSP90) and the AHR-interacting protein (AIP, formerly known as XAP2). Upon ligand binding, AhR dissociates from this complex and translocates to the nucleus, where it forms a heterodimer with the aryl hydrocarbon receptor nuclear translocator (ARNT). The AhR/ARNT complex binds to dioxin response elements (DREs) in DNA, initiating the transcription of target genes such as CYP1A1 and CYP1B1; created in BioRender.

**Figure 2 ijms-26-09160-f002:**
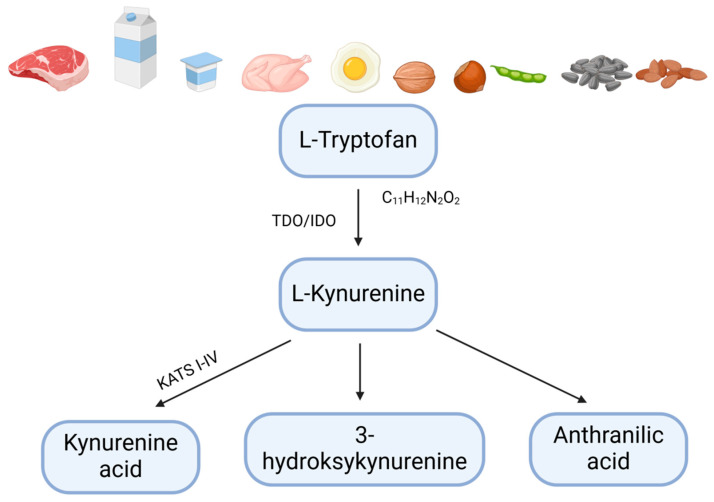
Tryptophan metabolism; created in BioRender.

**Figure 3 ijms-26-09160-f003:**
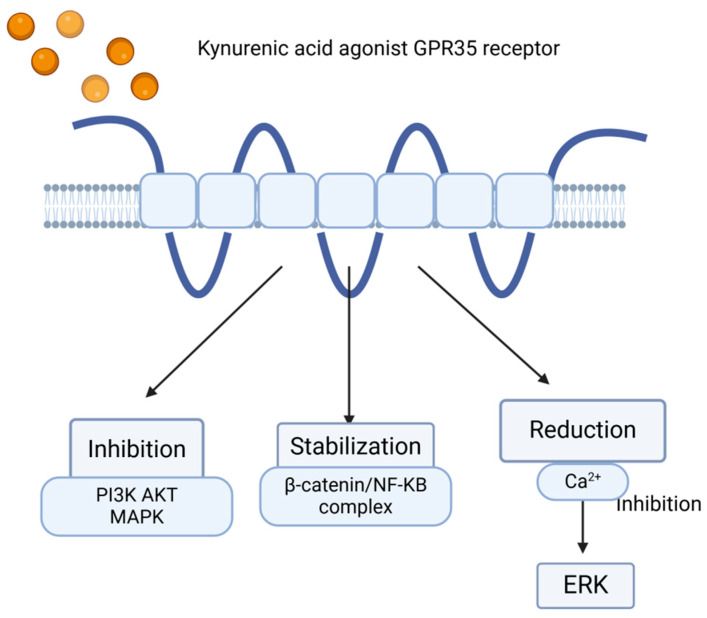
The KYNA–GPR35 interaction may lead to the suppression or attenuation of inflammation through the inhibition of PI3K, Akt, and MAPK, stabilization of the β-catenin/NF-κB complex, and reduction in Ca^2+^ levels, which consequently inhibits ERK kinase activity; created in BioRender.

## Data Availability

Data are available from the corresponding authors upon reasonable request.

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
