# Peer review of "Utilization of AhR and GPR35 Receptor Ligands as Superfoods in Cancer Prevention for Individuals with IBD"

_ijms, 2025, doi:10.3390/ijms26189160_

Round 1

Reviewer 1 Report

Comments and Suggestions for Authors

The authors here present a review of AhR and GPR53 ligands in cancer prevention for IBD.  This publication attempts to consolidate a large body of work in a review format.  Overall the information presented here seems accurate and prompts further questions about future work.  There are times at which references do not seem to align with the main discussion point of the text.  For example, the authors at times reference other large review studies and at times reference very specific in vitro studies.  The paper would be strengthened by looking critically at the references selected and finding stronger animal model or human based research to back some of the claims.

Text Comments:

  1. Please check that all references align with the text.  For example, the reader is led to believe reference 10 will discuss the side effects of anti-TNFα therapy based on the in-text context, but it is unrelated.

  2. Line 244: Reference 35 would be better replaced by one of the original papers describing the AHR-kynurenine-T cell axis (PMID: 20720200, PMCID: PMC2952546)
  3. Lines 214-219: There are many stronger studies in the literature on the role of AHR in DSS-induced colitis.  Consider for example (PMID: 27433903, PMCID: PMC5125557)

Figure Comments:

1. The paper would benefit from revising Figure 1 to ensure that text size and font are consistent and labels are aligned better.  In addition, AIP is the preferred name for XAP2 and will be better received in the AHR community.

Reviewer 2 Report

Comments and Suggestions for Authors

The authors present a narrative review on the potential of dietary ligands targeting the aryl hydrocarbon receptor and GPR35 as functional foods for managing inflammatory bowel disease and preventing colorectal cancer. The manuscript synthesizes evidence on the molecular pathways linking chronic inflammation, CYP metabolism, AhR/GPR35 signaling, and carcinogenesis. The emphasis on dietary ligands in the context of IBD-related carcinogenesis addresses a niche but increasingly relevant area of research. However, the current version largely reiterates known findings, lacks methodological transparency, overstates translational implications, and does not sufficiently reconcile contradictory evidence. While several mechanistic pathways are outlined, the level of critical evaluation is uneven, and some claims extend beyond the available human data. With a more rigorous and balanced synthesis, and clearer framing of the “superfood” concept, this review could make a valuable contribution.

Here are my detailed comments and suggestions:

  1. The manuscript attempts to cover multiple domains including inflammation, CYPs, AhR, GPR35, nutraceuticals, probiotics, and “superfoods”, but the connections among these elements are not always well integrated.
  2. No description is provided of how the reviewed studies were identified or selected, raising concerns about completeness and potential bias.
  3. Please include a clear methodology: databases searched, time frame, keywords, inclusion/exclusion criteria, and whether PRISMA or narrative review guidelines were followed.
  4. Clarify how the discussed nutraceuticals (e.g., KYNA, quercetin, curcumin, probiotics) were identified and prioritized. Was a systematic search performed?
  5. Provide quantitative comparisons between physiological dietary ligand levels and the concentrations required to activate AhR/GPR35 in humans.
  6. The review emphasizes chronic inflammation, but genetic, microbiome, and therapeutic confounders are under-discussed.
  7. CYP polymorphisms are mentioned, but their clinical significance in IBD-related CRC risk is not critically evaluated. Current discussion remains descriptive rather than analytical.
  8. The manuscript suggests that KYNA, quercetin, curcumin, and probiotics may prevent CRC in IBD patients, but most supporting evidence is preclinical. This risks overstating translational readiness.
  9. Claims should be tempered by distinguishing clearly between preclinical, clinical, and speculative evidence.
  10. Several AhR ligands, including KYNA and quercetin, exhibit both protective and carcinogenic effects depending on context. This duality is mentioned but not sufficiently analyzed.
  11. The review should expand its discussion of context-dependent effects, and provide a balanced synthesis rather than leaving contradictions unresolved.
  12. The translational implications of AhR ligands should include discussion of strategies to minimize potential carcinogenic risks in clinical application.
  13. The title refers to “superfoods”, but the manuscript does not operationalize this concept beyond listing nutraceuticals.
  14. Please define what qualifies as a “superfood” in a scientific context and apply the term consistently throughout.
  15. Related terms such as “chemopreventive” and “anti-cancer potential” are used loosely and should be more precisely defined or toned down to avoid overstatement.
  16. The conclusions are overstated. Reframe as hypothesis-generating, emphasizing knowledge gaps and research priorities rather than suggesting definitive clinical recommendations.
